# Single-Cell Heterogeneity of Epigenetic Factor Regulation Deciphers Alteration of RNA Metabolism During Proliferative SHH-Medulloblastoma

**DOI:** 10.3390/cancers17213424

**Published:** 2025-10-24

**Authors:** Raquel Francés, Jenny Bonifacio-Mundaca, Íñigo Casafont, Christophe Desterke, Jorge Mata-Garrido

**Affiliations:** 1Cell and Tissue Biology Group, Anatomy and Cell Biology Department, University of Cantabria-IDIVAL, 39011 Santander, Spain; rfrancesr@unican.es (R.F.); inigo.casafont@unican.es (Í.C.); 2National Tumor Bank, Department of Pathology, National Institute of Neoplastic Diseases, Surquillo 15038, Peru; jen-ny.bonifacio@upch.pe; 3Faculté de Médecine du Kremlin Bicêtre, Université Paris-Saclay, 94270 Le Kremlin-Bicêtre, France

**Keywords:** medulloblastoma, epigenetics, RNA metabolism, sc-RNAseq

## Abstract

Medulloblastoma is the most common malignant brain tumor in children and includes several molecular subtypes with distinct clinical outcomes. Because genetic mutations are relatively rare in this tumor, our study focused on understanding how epigenetic regulation—the chemical and structural changes that control gene activity without altering DNA—contributes to disease progression. Using data from large patient cohorts and single-cell sequencing, we identified specific epigenetic factors that distinguish tumor subtypes and correlate with prognosis. We also developed an epigenetic score that predicts patient survival and is particularly elevated in aggressive Sonic Hedgehog (SHH) tumors. These results suggest that dysregulated epigenetic programs affecting RNA metabolism and cell proliferation play a key role in medulloblastoma aggressiveness, highlighting new potential targets for therapy.

## 1. Introduction

Medulloblastoma is the most common malignant brain tumor in children, accounting for 25–30% of pediatric brain tumors and over 40% of childhood posterior fossa tumors [1]. It is classified into four distinct molecular subtypes, WNT, Sonic Hedgehog (SHH), Group 3, and Group 4, each with unique molecular and clinical characteristics [2]. These subtypes exhibit different responses to current multimodal therapies, which include surgical resection, craniospinal irradiation (for patients older than three years), and combination chemotherapy [3]. Among them, the WNT subtype has the most favorable prognosis, while Group 3 with TP53 mutations exhibits the poorest outcomes [4].

The mutational landscape of medulloblastoma is relatively sparse compared to other cancers. However, mutations affecting genes involved in DNA methylation, chromatin remodeling, and histone modification have been observed [5]. These include alterations in demethylases, acetyltransferases, and nucleosome remodelers, which further refine the classification of medulloblastoma molecular subtypes [5,6,7]. Moreover, aberrant patterns of histone H3 lysine methylation—particularly H3K4 and H3K27—are reported across subtypes [8]. DNA methylation also contributes to tumorigenesis by repressing specific genes, such as *PTCH1* (a negative regulator of the SHH signaling pathway), members of the *SFRP* family (inhibitors of the WNT pathway), and *ZIC2* (a transcriptional repressor).

Epigenomic analyses have revealed further heterogeneity within subtypes, identifying four epigenetic subgroups within SHH medulloblastoma and eight within Group 3 and Group 4, each associated with distinct clinical parameters [2,9,10,11].

Epigenetic factors are molecules that initiate or modulate epigenetic modifications, typically by altering DNA accessibility or recruiting other regulatory proteins [12]. To support research into these regulators, the EpiFactors database was developed. It catalogs a wide range of molecules involved in epigenetic control, including histones, histone variants, protamines, histone chaperones, histone modifiers, readers of histone modifications, chromatin remodelers, DNA/RNA modifiers and their readers, and protein cofactors that interact with epigenetic complexes essential for their function [13].

Recent studies have underscored that both DNA and histone methylation critically influence medulloblastoma biology. Aberrant hypermethylation of tumor suppressor genes (e.g., *PTCH1*, *CDKN2A*, and *SFRP* family members) promotes oncogenic SHH and WNT signaling activation, while global hypomethylation favors genomic instability and enhancer activation [14]. In parallel, alterations in histone methylation marks such as H3K27me3 and H3K4me3 modulate differentiation and proliferation programs in a subgroup-specific manner. For example, loss of H3K27me3 has been associated with Group 3/4 tumors displaying stem-like transcriptional states, whereas aberrant H3K4 methylation contributes to SHH-MB tumor growth through the activation of GLI-dependent networks [10,14]. Together, these findings support that coordinated changes in DNA and histone methylation shape the epigenetic landscape and aggressiveness of medulloblastoma.

In this study, we investigated the expression patterns of epigenetic factors in medulloblastoma using bulk and single-cell transcriptomic data. We found that members of the SWI/SNF superfamily were dysregulated across all four molecular subtypes. Subtype-specific alterations were also identified: for example, the acetyltransferase complex was shared between Group 3 (with *SMARCD3* as a marker) and Group 4 (with *RBM24* as a marker); *SWR1*, β-catenin/TCF, and protein–DNA complexes were enriched in SHH-MB (with *EYA1* and *SATB2* as potential markers); and RSC-type, PRC1, DNA polymerase complexes, and X-chromosome-related elements were enriched in WNT-MB (with *FOXA1* and *PIWIL4* as markers). Additionally, we developed an epigenetic score (epi-score) associated with RNA metabolism and S-adenosyl-L-methionine activity, which emerged as an independent adverse prognostic factor. High epi-scores were observed in proliferative, stem-like SHH malignant cells (characterized by G2/M cell cycle phase, low pseudotime, and high entropy) exhibiting de-regulation of RNA splicing, DNA recombination, and nuclear division.

Overall, our findings suggest that expression heterogeneity of epigenetic factors is intricately linked to the molecular classification and prognosis of medulloblastoma.

## 2. Materials and Methods

### 2.1. The Open Pediatric Brain Tumor Atlas (PBTA) [15]

RNA sequencing data for medulloblastoma from the PBTA were obtained via the Pediatric cBioPortal platform, part of the “Open Pediatric Cancer Project” consortium [16], accessible at: https://pedcbioportal.kidsfirstdrc.org/ (accessed on 6 November 2024). After quality control filtering and clinical annotation, the cohort included 254 tumor samples. The majority of samples were derived from male patients (61%) and from primary tumors (61%). Molecular subgroup distribution was as follows: WNT (*n* = 21), SHH (*n* = 74), Group 3 (*n* = 60), and Group 4 (*n* = 99) (Table 1). RNA-seq count data were normalized using a log2(count + 1) transformation. Batch effects were corrected using the ComBat algorithm [17].

### 2.2. Williamson Cohort (RNA-seq)

An independent cohort of medulloblastoma RNA sequencing data was downloaded from the ArrayExpress database under accession number E-MTAB-10767 [18]. This dataset comprised 331 RNA sequencing experiments, with most tumor samples derived from male pediatric patients. All four molecular subtypes were represented in this validation cohort; however, overall survival follow-up data were available only for patients classified as Group 3 (G3) and Group 4 (G4). For preprocessing, RNA-seq quantifications were transformed into pseudo-counts, normalized using quantile normalization, and batch effects were corrected using the ComBat algorithm [17].

### 2.3. sc-RNA Sequencing of Medulloblastoma Tumors (n = 28)

The GSE155446 dataset consists of single-cell RNA sequencing data from 28 pediatric medulloblastoma samples, representing all four molecular subtypes: SHH, WNT, Group 3 (G3), and Group 4 (G4). Libraries were prepared using Chromium Single Cell V2 and V3 Chemistry Library Kits (10× Genomics), and barcoded cDNA libraries were sequenced on an Illumina (San Diego, CA, USA) NovaSeq 6000 platform, achieving a sequencing depth of approximately 50,000 reads per cell. Demultiplexing and initial processing were performed using CellRanger (10× Genomics) (GSE155446) [19]. Raw count matrices, corrected for batch effects, were integrated into a single Seurat object using Seurat version 5.1.0. Standard preprocessing steps—including NormalizeData, FindVariableFeatures, ScaleData, and RunPCA—were applied following the Seurat pipeline [20]. A single-cell expression score was calculated using the epifactor gene signature via the AddModuleScore function in Seurat.

### 2.4. Bulk Transcriptome Analysis

All bioinformatics analyses were performed using the R software environment (v4.4.2). To perform leave-one-out supervised machine learning for molecular subgroup classification, we used the pamr R package (v1.57) [21], applying it to the full transcriptome restricted to the expression of human epigenetic regulators obtained from the EpiFactors database (v2) [13]. Briefly, the leave-one-out procedure with the pamr (nearest-shrunken centroids) algorithm is a supervised classification approach in which each sample is in turn held out as a test case while a predictive centroid-based model is trained on the remaining samples; cross-validation produces an estimate of the classifier error rate and identifies a minimal set of predictive features. In our case this test evaluates whether the expression of epigenetic regulators alone can robustly discriminate canonical medulloblastoma subgroups—a low cross-validated error (4% in PBTA; 8% in Williamson) indicates that the epigenetic signature has strong subgroup specificity [21].

Transcriptome-level analyses—including principal component analysis (PCA) and visualization (e.g., expression heatmaps)—were conducted using the transpipe R package (v1.4), available at: https://github.com/cdesterke/transpipe14 (accessed on 22 March 2025). Experimental batch effects in transcriptome datasets were identified using *k*-means clustering and corrected using the ComBat function implemented in the sva R package (v3.54) [17].

Functional enrichment analysis of transcriptomic gene lists was performed using the clusterProfiler R package (v4) [22], with annotations from the Gene Ontology (GO) database [23]. Protein–protein interaction (PPI) networks were constructed using the STRING online platform [24], with functional enrichment based on Gene Ontology [23] and UniProt databases [25].

Of note, classical oncogenes and tumor suppressors such as *MYC* and *TP53* were not included in the EpiFactors v2 database, which specifically annotates molecules directly involved in epigenetic regulation (e.g., histone/DNA/RNA modifiers, chromatin remodelers, and readers) [13]. Consequently, these genes were absent from our input list and were not considered as predictive epifactors. Nevertheless, *MYC* and *TP53* expression remained positively correlated with the high epifactor score, supporting that the epi-score captures biological programs active in *MYC*-driven and *TP53*-mutant medulloblastomas.

### 2.5. Gene Expression Survival Analyses

Univariate Cox overall survival analyses against the expression of predictive epifactors were performed with loopcolcox v1.0.0 R-package available at the following address: https://github.com/cdesterke/loopcolcox (accessed on 22 March 2025) [26]. The score of metabolism/mitochondria enrichment was computed according to the following formula:score = Σ exp × beta,(1)
in which “exp” is the value of gene expression and “beta” it’s the corresponding Cox overall survival beta coefficient. The Cox beta value for individual epifactors was obtained with loopcolcox R-package. Kaplan–Meier and log rank tests were performed with survival (v3.8-3) [27] and survminer (v0.5.0) R-packages. Multi-variable overall survival model was built with “coxph” function from the survival package and bootstrapping calibration of the model was performed using the RMS R-package v7.0-0 [28]; a nomogram of the calibrated multi-variable model was drawn with regplot R-package v1.1. For transparency and reproducibility, the proportional hazards assumption was formally tested using Schoenfeld residuals with the cox.zph function from the survival R package (non-significant global and variable-specific tests indicate no time-dependent effects). Harrell’s concordance index (C-index) was computed to quantify the model’s discriminative ability (C-index = 0.5 indicates random prediction; values ≥ 0.70 are usually interpreted as good discrimination). Bootstrap calibration of the multi-variable model was performed using the rms::calibrate function with 1000 resampling steps to obtain an optimism-corrected estimate of predicted versus observed survival probability at the 25-month time point. These analyses were implemented in R (v4.4.2) [28].

### 2.6. Single-Cell RNA Sequencing Analyses

Original single-cell RNA sequencing data [19] were processed as Seurat objects using the Seurat R package (v5) [20]. Standard preprocessing steps—including NormalizeData, FindVariableFeatures, ScaleData, and RunPCA—were applied to each object. Visualization of single-cell data was performed using various Seurat functions, including FeaturePlot, DimPlot, VlnPlot, and DotPlot. Cell cycle phase prediction was also conducted using the Seurat package.

A single-cell epifactor score was computed based on the expression of predictive molecules using the AddModuleScore function in Seurat. For malignant SHH medulloblastoma cells, the Seurat object was converted into a SingleCellExperiment object [29], and cell entropy was calculated using the TSCAN R package (v1.44.0) [30,31]. Pseudotime trajectories were computed using the slingshot R package (v2.14.0) [32]. Pseudotime is an inferred ordering of single cells along a putative developmental or differentiation trajectory based on transcriptional similarity; it does not measure chronological time but rather a relative progression (early → late) in cell-state space (Slingshot implements this by fitting lineages in reduced-dimension space). Cellular entropy (computed here with the TSCAN package) measures transcriptional heterogeneity at the single-cell level: higher entropy indicates a broader or more “promiscuous” transcriptional program, often associated with stem-like or plastic cell states. Together, low pseudotime (early location on the trajectory) and high entropy suggest cells that are less differentiated and more transcriptionally plastic—properties linked to proliferative/stem-like behavior and, in cancer, to aggressive phenotypes and therapy resistance [32].

## 3. Results

### 3.1. Epigenetic Factor Regulation Reflects Molecular Subtypes in Medulloblastoma

#### 3.1.1. Epigenetic Factor Expression in the PBTA Bulk Transcriptome Cohort

The medulloblastoma transcriptome cohort was obtained from the PBTA portal (Table 1), quantile normalized (Appendix A), and batch-corrected using the ComBat algorithm following k-means cluster detection (Appendix A). This correction enabled the clear stratification of samples according to their molecular subtypes (Appendix A).

Expression data for epigenetic regulators were extracted from the normalized transcriptome and used to perform a leave-one-out supervised machine learning analysis based on molecular subtype classification. A minimal predictive signature comprising 64 epigenetic factors (epifactors) was identified through cross-validation (Figure 1A; Appendix A). This signature achieved a global classification error rate of only 4% (Figure 1B), with strong subgroup-specific accuracy during cross-validation (Figure 1C).

Hierarchical clustering (Euclidean distance with Ward.D2 method) using the 64-epifactor signature effectively grouped samples by molecular subtype (Figure 1D). Additionally, unsupervised principal component analysis (PCA) based on this signature confirmed clear subtype separation (Figure 1E).

#### 3.1.2. Functional Characterization and Validation of Subtype-Specific Epigenetic Signatures

Functional enrichment analysis of subtype-specific epifactors (Appendix A) revealed distinct biological processes based on Gene Ontology (GO) terms (Figure 2A). Epifactors specific to Group 3 and Group 4 shared enrichment in several chromatin-modifying complexes, notably the histone acetyltransferase (HAT) complex (Figure 2B). In Group 3, this included HAT1, ELP3, KAT2A/GCN5, and SUPT3H, whereas Group 4 enrichment involved KAT5/TIP60 and TAF9. Although SMARCD3 (BAF60C) and RBM24 are not acetyltransferases, they were co-expressed with these HATs and may participate in associated chromatin-remodeling or RNA-binding assemblies [33]. The SWI/SNF superfamily complex appeared enriched across all four molecular subtypes (Figure 2A).

Group 3 epifactors were specifically associated with enrichment of the Sin3-type complex and methyltransferase complex (Figure 2A,B). In Group 4, a unique enrichment for the ATAC (Ada Two A–containing histone acetyltransferase) complex was observed (Figure 2A,B). SHH-specific epifactors were enriched in the Swr1 complex, β-catenin/TCF complex, and protein–DNA interaction complexes (Figure 2A,B). In the WNT subtype, enrichment was identified in X-chromosome-related factors, the RSC-type complex, PRC1 complex, and DNA polymerase complex (Figure 2A,B).

Several individual epifactors were also highlighted for their strong predictive value. *EYA1* and *SATB2* were among the top predictors for the SHH subtype, *FOXA1* and *PIWIL4* for the WNT subtype, *RBM24* for Group 4, and *SMARCD3* for Group 3 (Appendix A).

**Figure 1 cancers-17-03424-f001:**
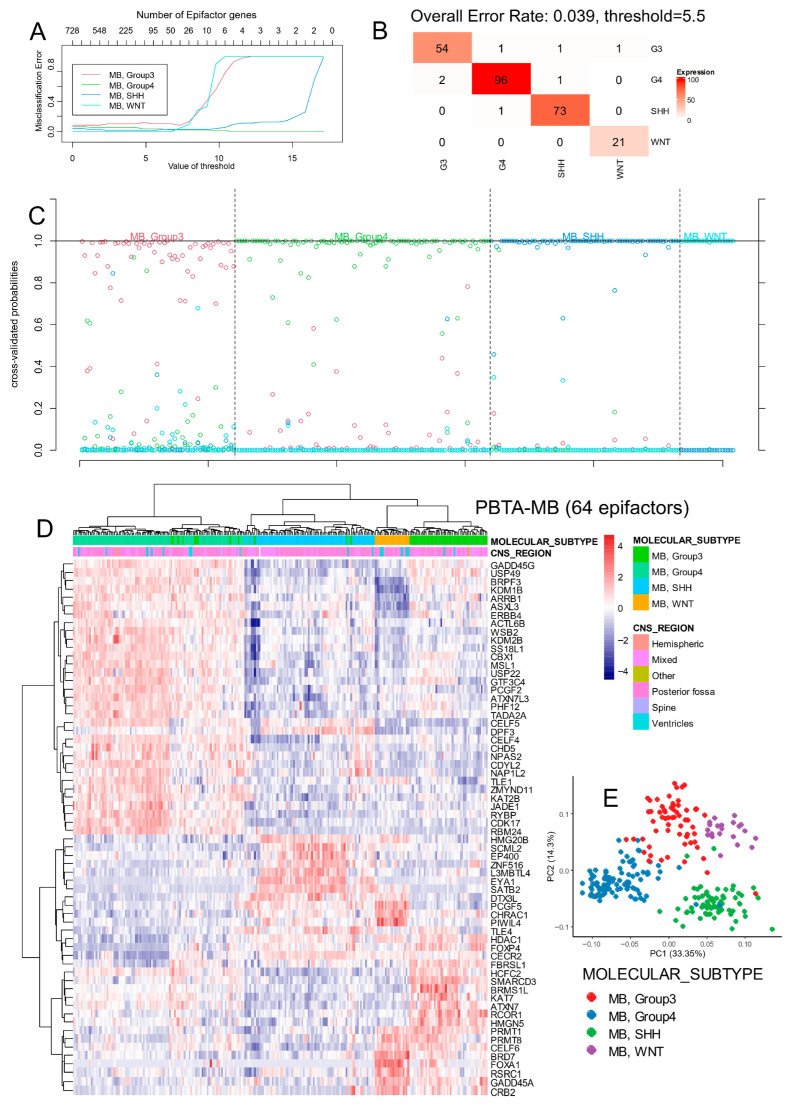
Stratification of medulloblastoma molecular subtypes with expression of 64 epifactors in PBTA cohort. (**A**) Misclassification plot based on expression of epifactors for stratification of four molecular subtypes of medulloblastoma (Group3, Group4, WNT, and SHH). (**B**) Confusion matrix supervised on molecular subtype classification based on expression of 64 predictive epifactors. (**C**) Cross-validation probability plot for individual samples. (**D**) Unsupervised clustering (Euclidean distances) and expression heatmap based on expression of 64 predictive epifactors. (**E**) Principal component analysis based on expression of 64 predictive epifactors.

**Figure 2 cancers-17-03424-f002:**
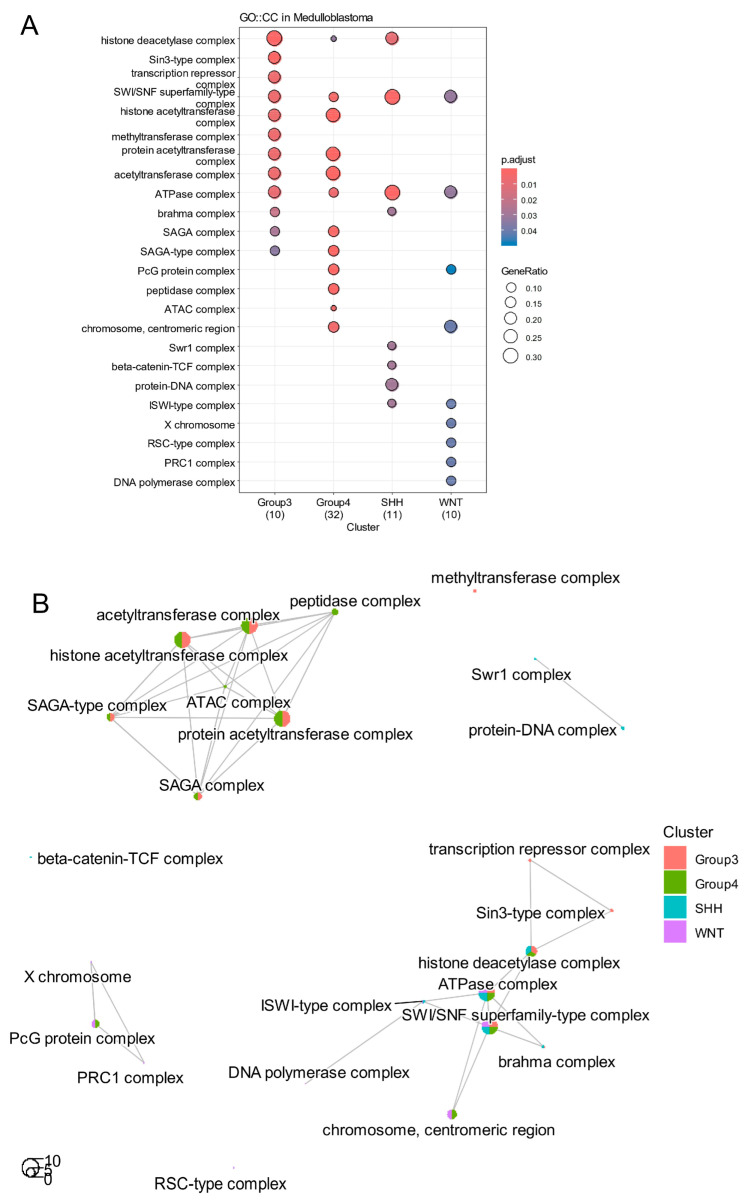
Heterogeneity of epigenetic activation across distinct molecular subtypes of medulloblastoma. (**A**) Dot plot of functional enrichment performed with epifactors predictive of each molecular subtypes on Gene Ontology Cellular Component database (GO::CC). (**B**) Functional enrichment network performed with epifactors predictive of molecular subtypes during medulloblastoma.

#### 3.1.3. Epigenetic Factor Regulation in the Williamson Bulk Transcriptome Cohort

Following ComBat batch correction of the Williamson RNA sequencing cohort (Table 2; Appendix A), 62 of the 64 previously identified predictive epifactors (Appendix A) were found to be expressed in this independent dataset. A leave-one-out supervised machine learning analysis using these 62 epifactors successfully discriminated molecular subtypes in this validation cohort (Figure 3A), achieving an overall classification error rate of 8% (Figure 3B). Subtype-specific stratification of individual tumor samples was robust following cross-validation (Figure 3C).

Hierarchical clustering based on the expression of the predictive epifactors (using Euclidean distance and Ward.D2 method) grouped the majority of samples correctly according to molecular subtype (Figure 3D). Principal component analysis further confirmed this clustering (Figure 3E).

Subtype-specific markers identified in the PBTA cohort were validated in this independent cohort. *PIWIL4* remained a top predictor for the WNT subtype; *EYA1* and *SATB2* were again among the strongest predictors for SHH; *SMARCD3* for Group 3; and *RBM24* for Group 4 (Appendix A).

### 3.2. Epifactor Expression Score Related to RNA Metabolism Is Associated with Medulloblastoma Prognosis

Using the PBTA bulk transcriptome cohort (Table 1), we performed a Cox proportional hazards survival analysis for each epigenetic factor’s expression. A subset of epifactors (*n* = 29) was found to be significantly and positively associated with poor overall survival in medulloblastoma patients (highlighted as orange dots in Figure 4A; see details in Table 3).

Using the 29 epigenetic factors found to be significantly associated with adverse prognosis (Table 3; Appendix A), a protein–protein interaction (PPI) network was constructed using the STRING database with default parameters (Figure 4B). Functional enrichment analysis of this network, based on the Gene Ontology (GO) database, revealed that 16 of these epifactors were involved in RNA metabolism processes (highlighted in blue, Figure 4B). In addition, four epifactors were found to be functionally associated with S-adenosyl-L-methionine metabolism (highlighted in red, Figure 4B), according to UniProt-based annotation.

An epifactor expression score was calculated by weighting the expression levels of these 29 predictive epifactors by their Cox regression beta coefficients (Table 3). Stratifying patients in the PBTA cohort based on this score revealed a significant association with poor prognosis for those with a high epifactor score compared to those with a lower score (log-rank *p* < 0.0001, Figure 4C). The expression of these 29 epifactors also enabled the separation of molecular subtypes via hierarchical clustering and principal component analysis (Figure 4D,E).

Within the PBTA cohort (Table 1), patients with high epifactor scores were significantly enriched for the following: —Female sex (48% vs. 28%, *p* = 0.0025);—SHH subtype (46% vs. 10%, *p* < 1 × 10^−4^);—Group 3 subtype (30% vs. 15%, *p* < 1 × 10^−4^);—Metastatic tumors (38% vs. 27%, *p* = 0.04);—Recurrent tumors (6.5% vs. 1.7%, *p* = 0.04).

To validate these findings, the epifactor expression score was also computed in the Williamson cohort, and stratification into low- and high-score groups again confirmed a significant association with poor prognosis (log-rank *p* < 1 × 10^−4^; Figure 4F). In this cohort, patients with high scores were also enriched for the following:—SHH and Group 3 subtypes (*p* < 2.22 × 10^−19^; Table 2; Figure 4G);—Large cell/anaplastic histology (*p* = 0.0027; Table 2);—TP53 mutations (*p* < 1 × 10^−4^; Table 2);—hTERT promoter mutations (*p* = 0.00037; Table 2);—MYC amplifications (*p* = 0.000298; Table 2);—MYCN amplifications (*p* = 0.00062; Table 2).

Unsupervised hierarchical clustering based on these epifactor expressions also showed a trend toward the reclassification of molecular subtypes in the Williamson dataset (Figure 4H). Consistently, high-score tumors exhibited elevated *MYC* and *TP53* expression, in agreement with the enrichment of *MYC* amplification and *TP53* mutation reported in Table 2.

To verify model assumptions and performance, we (i) tested the proportional hazards assumption using Schoenfeld residuals (no significant violations were observed; see Appendix A), (ii) quantified discrimination with Harrell’s concordance index (C-index > 0.75, indicating good separation of low- and high-risk cases), and (iii) assessed calibration by bootstrap (1000 resamples) of predicted 25-month survival against observed outcomes (Figure 5B). Together these diagnostics indicate that the epifactor expression score contributes independent prognostic information beyond routine clinical covariates and is robustly predictive in the PBTA cohort. Biologically, this supports the hypothesis that epigenetic programs captured by the epi-score reflect aggressive tumor biology (higher prevalence of metastatic and recurrent cases and association with SHH/Group 3 and TP53/MYC alterations—see Table 1 and Table 2).

The epifactor expression score was confirmed as an independent adverse prognostic factor in the multi-variable model (*p* = 6.69 × 10^−4^; Table 4). To assess predictive calibration, a bootstrap calibration curve was generated at the 25-month failure time point (Figure 5B). The corresponding nomogram demonstrated the substantial contribution of the epifactor score to the prediction of overall survival at this time frame (Figure 5C).

### 3.3. Epifactor Expression Score Is Elevated in Malignant Cells from the SHH Medulloblastoma Subtype at the Single-Cell Level

To investigate the heterogeneity and expression of prognostic epifactors at single-cell resolution in medulloblastoma, we analyzed single-cell RNA sequencing data from the GSE155446 dataset, comprising 34,243 tumor-associated cells. This dataset includes samples from 28 patients (Figure 6A), representing all four molecular subtypes: Group 3, Group 4, SHH, and WNT (Figure 6B).

The dataset is composed predominantly of malignant cells, along with a minority of non-tumor cell types including lymphocytes, monocytes/macrophages, oligodendrocytes, and astrocytes (Figure 6C).

An epifactor expression score was computed at the single-cell level based on the expression of the 29 adverse prognostic epifactors (Table 3). A subset of malignant cells showed a positive epifactor score, indicating activation of the adverse transcriptional program (Figure 6D).

Several individual epifactors demonstrated subtype-specific expression patterns among malignant cells (Figure 6E):—*BAZ1A* was highly expressed in malignant cells from the SHH subtype.—*FBL* showed widespread expression in WNT and SHH subtypes.—*UBE2D3* was particularly enriched in malignant cells from Group 4.—*SYNCRIP* was highly expressed in malignant cells from Group 3.

Violin plot analysis of malignant cells confirmed that SHH subtypemalignant cells had a significantly higher epifactor expression score compared to other molecular subtypes (Figure 6F; Table 5), indicating a potential link between the SHH subtype and heightened adverse epigenetic transcriptional activity at the single-cell level.

### 3.4. SHH Medulloblastoma Malignant Cells Exhibit Elevated Epifactor Scores in Proliferative States with High Cellular Entropy

Single-cell analyses were further focused on malignant cells from SHH medulloblastoma patients (Figure 7A). The concordance of high epifactor score, G2/M cell cycle state, low pseudotime (early position on the inferred lineage), and high cellular entropy is consistent with a proliferative, stem-like malignant population; such a population is biologically relevant because stem-like/proliferative cells have been implicated in medulloblastoma growth, dissemination, and treatment resistance. Uniform Manifold Approximation and Projection (UMAP) revealed that these cells were spatially organized according to their cell cycle phase, as inferred from computational phase prediction (Figure 7B).

The expression of individual adverse epifactors was found to be heterogeneous across cell cycle phases (Figure 7C), specifically the following:—BAZ1A and FBL were predominantly expressed during the S phase;—UBE2D3 was mainly expressed in G1 phase;—SYNCRIP, PTBP1, and TAF9 were up-regulated during the G2/M phase.

Computation of the epifactor expression score at the single-cell level revealed that cells with high scores localized primarily on the left side of the UMAP, which corresponded to the G2/M phase cluster (Figure 7B,D). Violin plot analysis confirmed that epifactor expression scores were significantly higher in G2/M-phase malignant cells (Figure 7E).

Moreover, these G2/M-phase cells also exhibited higher cellular entropy (Figure 7F), indicating greater transcriptional diversity and potentially a more plastic or stem-like state.

To investigate the developmental trajectory of these cells, pseudotime analysis was performed using the Slingshot algorithm after downsampling the dataset to maintain the spatial and phase-based structure (Figure 7G). Slingshot inferred a unidirectional pseudotime trajectory in SHH malignant cells (Figure 7H), consistent with patterns of cell cycle phase, epifactor score, and entropy (Figure 7I).

Importantly, cells with the highest epifactor expression scores were located at the early pseudotime (Figure 7J), aligning with G2/M-phase cells and those with the highest cellular entropy (Figure 7K), suggesting that these cells may represent an early-stage, stem-like malignant population.

Functional enrichment analysis of the genes associated with the pseudotime trajectory (Appendix A) revealed significant involvement in RNA metabolism, RNA splicing, and negative regulation of transcription (Figure 8A–C), with additional enrichment for DNA recombination and nuclear division, supporting the notion of active proliferative and transcriptional regulation in these cells.

## 4. Discussion

In this study, we demonstrate that SWI/SNF superfamily alterations are present across all four medulloblastoma (MB) molecular subtypes. SWI/SNF complexes are ATP-dependent chromatin remodeling machineries that regulate DNA accessibility for transcription factors and other regulatory proteins [34]. By repositioning nucleosomes, SWI/SNF complexes modulate gene expression and contribute to neuronal development [35]. These complexes are also essential for enhancer-driven transcriptional activation, particularly during neurogenesis [14,36]. Several SWI/SNF components, such as *ARID1A*, *ARID2*, and *SMARCA4*, are known to harbor mutations in MB tumors [5,6,7].

Our findings also indicate that specific epigenetic complexes are selectively de-regulated in certain MB subtypes. For instance, the acetyltransferase complex appears to be affected in Group 3 and Group 4 tumors, with *SMARCD3* serving as a Group 3 marker and *RBM24* as a Group 4 marker. Histone acetyltransferases (HATs) loosen the chromatin structure, promoting gene transcription and playing critical roles in brain development. For example, the down-regulation of hMOF (KAT8), a histone H4K16 acetyltransferase, has been linked to poor prognosis in medulloblastoma [37]. In our data, Group 3 tumors with SMARCD3 (BAF60C) up-regulation and Group 4 tumors with RBM24 up-regulation also exhibited reduced hMOF expression, suggesting a compensatory or alternative chromatin-regulatory mechanism rather than a direct enzymatic overlap. Although SMARCD3 and RBM24 are not acetyltransferases, they were co-expressed with members of the HAT complex (e.g., HAT1, KAT2A/GCN5, and KAT5/TIP60) and may participate in associated chromatin-remodeling or RNA-binding assemblies [33]. SMARCD3 influences DAB1-mediated Reelin signaling, which is crucial for Purkinje-cell migration and MB metastasis, by orchestrating cis-regulatory elements [38,39]. Moreover, RBM24 has previously been identified as part of a minimal six-gene signature predicting MB subtypes [40].

The SHH subtype was associated with SWR1-type nucleosome remodelers, β-catenin, *TCF*, and chromatin regulators such as *EYA1* and *SATB2*. The SWR1 complex facilitates histone exchange, specifically replacing canonical H2A with H2A.Z [41,42,43]. *EYA1* acts as a phosphatase promoting SHH signaling through cooperation with *SIX1*, enhancing *GLI* transcriptional activity. It plays a critical role in hindbrain development and SHH-MB growth [44]. *SATB2*, a member of the special AT-rich binding protein family, is a master chromatin organizer and has been implicated in neuronal identity and congenital craniofacial malformations [45,46,47,48].

The WNT subtype displayed alterations in RSC-type remodelers, PRC1, DNA polymerase complexes, and X-chromosome-related pathways, along with specific markers *FOXA1* and *PIWIL4*. *PIWIL4* has been shown to promote neuronal differentiation in stem cell models and modulate glioma-related factors such as *PTN* and *NLGN3* [49].

A core finding of this study is the identification of an epifactor expression score, derived from 29 adverse prognostic epigenetic regulators. This score was significantly associated with poor overall survival and was validated in two independent bulk transcriptome cohorts (PBTA and Williamson). Functional enrichment analysis revealed strong connections to RNA metabolism and S-adenosyl-L-methionine (SAM)-related pathways. SAM acts as a universal methyl donor in methylation processes involving DNA, RNA, proteins, and lipids [50]. In particular, RNA methylation, largely mediated by METTL3 and METTL14, influences mRNA splicing, stability, and translation, and plays a critical role in oncogenesis [51,52].

Among the epifactors contributing to the S-adenosyl-L-methionine (SAM)-linked module, two histone methyltransferases—SETD7 and PRDM13—were particularly enriched. SETD7 catalyzes H3K4me1 deposition and methylation of non-histone targets such as TP53 and RB1, thereby modulating cell cycle control and apoptosis [53]. In medulloblastoma, aberrant SETD7 activity may influence *TP53* signaling and the transcriptional stability of MYC-driven networks. PRDM13, a neuron-specific PR/SET protein, regulates GABAergic vs. glutamatergic fate during cerebellar development and is frequently silenced by promoter methylation in embryonal brain tumors [54]. The enrichment of both enzymes supports a model in which dysregulated histone and non-histone methylation contribute to defective neuronal differentiation and malignant transformation in MB.

High epifactor scores were predominantly found in Group 3 and SHH MB tumors, especially those harboring TP53 mutations or MYC/MYCN amplifications—known adverse prognostic features with <50% five-year progression-free survival [2,55].

At the single-cell level, malignant SHH cells with high epifactor scores were localized to the G2/M phase, characterized by low pseudotime and high entropy, indicative of stem-like properties. These cells also showed the up-regulation of genes involved in RNA splicing, DNA recombination, and nuclear division, processes that are tightly regulated during development but often hijacked in cancer [56]. These results reinforce the notion that epigenetic de-regulation plays a central role in tumor progression, heterogeneity, and treatment resistance in MB. Collectively, the epifactors identified here converge on processes that are crucial to medulloblastoma onset and progression. Members of the SWI/SNF and HAT complexes (e.g., SMARCD3, ELP3, KAT5) modulate enhancer accessibility and neuronal gene expression, while RNA-associated factors (PTBP1, SYNCRIP, FBL) regulate alternative splicing and ribosome biogenesis, thereby sustaining rapid proliferation. DNA damage and chromatin integrity regulators (BAZ1A, RCC1, USP12) further integrate transcriptional and replicative stress responses. These functions align with the concept that medulloblastoma cells exploit epigenetic plasticity to maintain a progenitor-like state. Consequently, pharmacologic targeting of histone acetylation (e.g., HDAC inhibitors) or methylation pathways (e.g., SAM-cycle modulators or SETD7 inhibitors) represents a plausible therapeutic avenue deserving experimental validation.

Despite its integrative design, this study has some limitations. First, our analyses are based on publicly available transcriptomic datasets; thus, validation using methylation or proteomic data was not possible. Second, while the epi-score was validated across independent cohorts, experimental functional validation of candidate epifactors in cellular or animal models remains necessary. Third, single-cell RNA-seq analyses may be biased by cell dissociation protocols or sequencing depth, potentially underrepresenting certain tumor or stromal populations. Future studies integrating DNA methylation, chromatin accessibility, and proteomics data will be essential to confirm the mechanistic role of these epigenetic alterations in medulloblastoma pathogenesis.

## 5. Conclusions

Our findings demonstrate that heterogeneity in epigenetic factor expression is strongly associated with both the molecular classification and prognosis of medulloblastoma (Figure 9). We identify a robust epifactor expression score linked to poor clinical outcomes, particularly in high-risk subgroups such as SHH and Group 3 MB. Furthermore, single-cell analysis uncovers a malignant subpopulation of SHH MB cells with high epifactor scores, proliferative potential, and stem-like features, suggesting epigenetic mechanisms as key therapeutic targets.

## Figures and Tables

**Figure 3 cancers-17-03424-f003:**
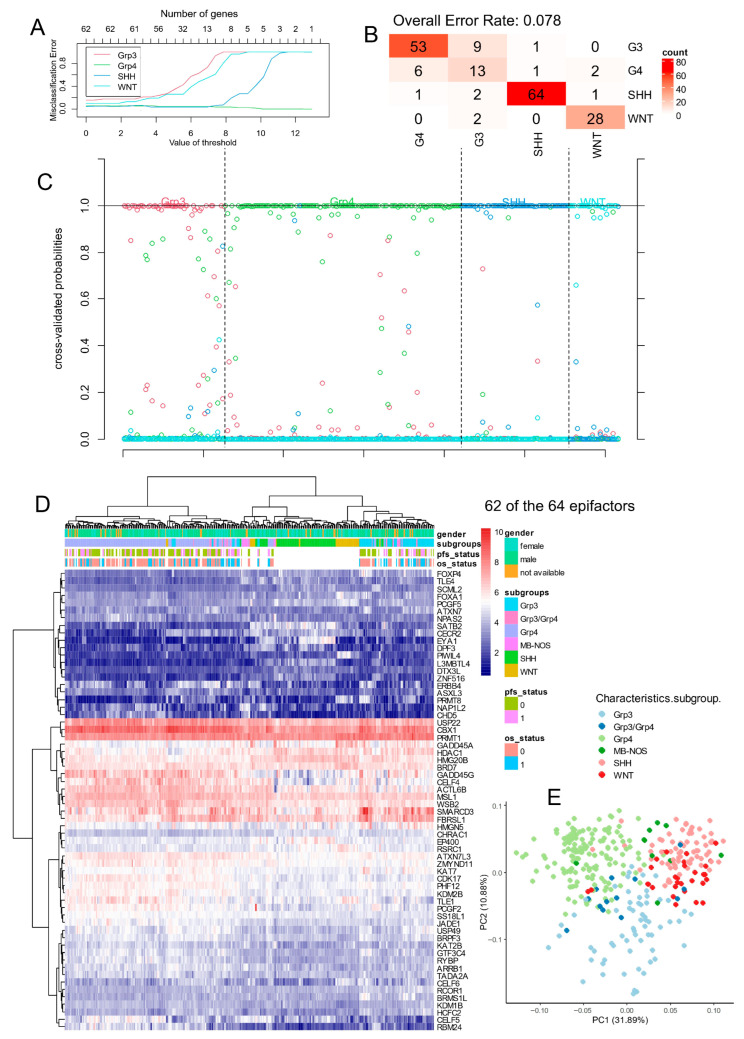
Stratification of medulloblastoma molecular subtypes with expression of 64 epifactors in Williamson cohort. (**A**) Misclassification plot based on expression of epifactors for stratification of four molecular subtypes of medulloblastoma (Group3, Group4, WNT, and SHH). (**B**) Confusion matrix supervised on molecular subtype classification based on expression of epifactor signature. (**C**) Cross-validation probability plot for individual samples. (**D**) Unsupervised clustering (Euclidean distances) and expression heatmap based on expression of epifactor signature. (**E**) Principal component analysis based on expression of epifactor signature.

**Figure 4 cancers-17-03424-f004:**
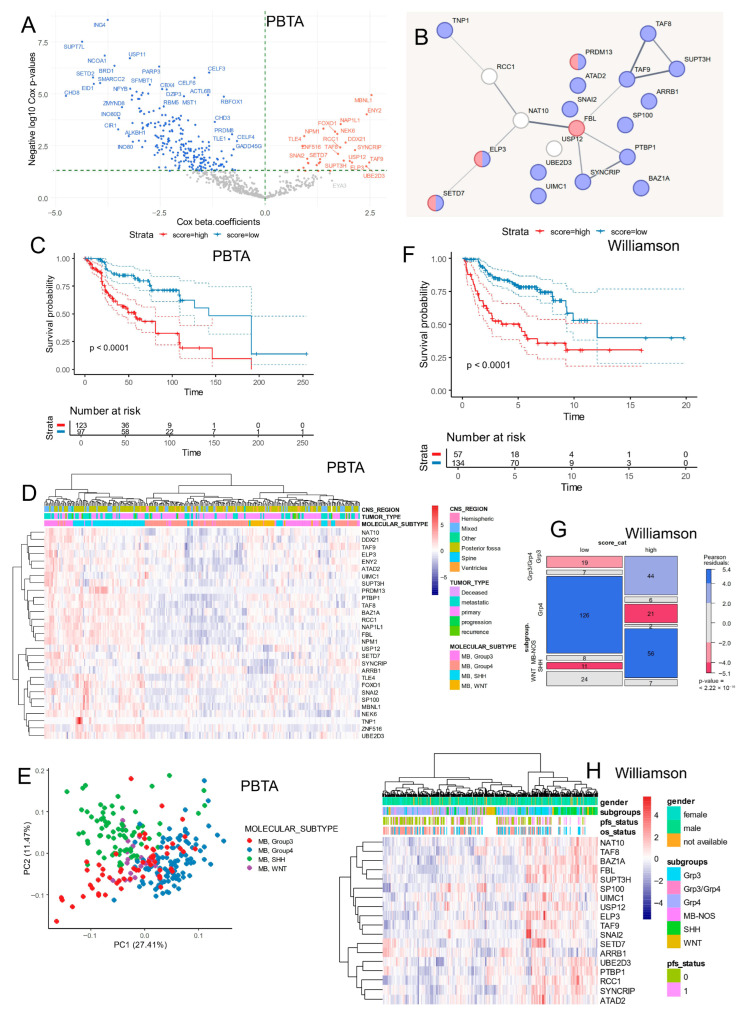
Implication of RNA metabolism in epifactor signature during bad prognosis. (**A**) Volcano plot of univariate Cox analyses according to epifactor expression and overall survival of medulloblastoma patients in PBTA cohort. (**B**) Protein–protein interaction network built with epifactors found with adverse expression during medulloblastoma (blue bubbles: regulation of RNA metabolism GO::BP FDR = 5.47 × 10^−5^; red bubbles: S-adenosyl-L-methionine Uniprot FDR = 0.048). (**C**) Kaplan–Meier and log rank test on epifactor score in PBTA medulloblastoma cohort. (**D**) Unsupervised clustering and expression heatmap of adverse epifactors in RNAseq PBTA cohort. (**E**) Principal component analysis based on expression of adverse epifactors in PBTA cohort. (**F**) Kaplan–Meier and log rank test on epifactor score in Williamson medulloblastoma cohort. (**G**) Mosaic plot testing association between epifactor score categories (low and high) versus molecular classification in medulloblastoma Williamson cohort. (**H**) Unsupervised clustering and expression heatmap of adverse epifactors in RNAseq Williamson cohort.

**Figure 5 cancers-17-03424-f005:**
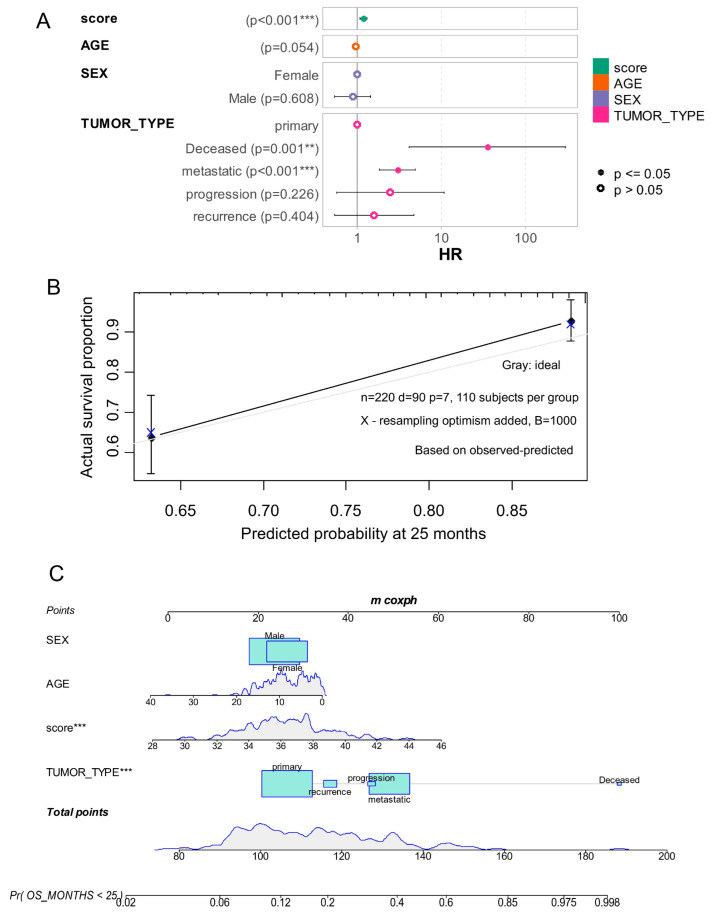
Epifactor expression score is adverse independent parameter in prognosis of medulloblastoma. (**A**) Forest plot of multi-variable overall survival model including epifactor score (score), age, sex and tumor types as covariates. (**B**) Calibration of multi-variable overall survival model by bootstrapping with 1000 resampling steps at 25 months of failure. (**C**) Nomogram of calibrated multi-variable overall survival model at 25 months of time failure. ** *p* = 0.001; *** *p* < 0.001.

**Figure 6 cancers-17-03424-f006:**
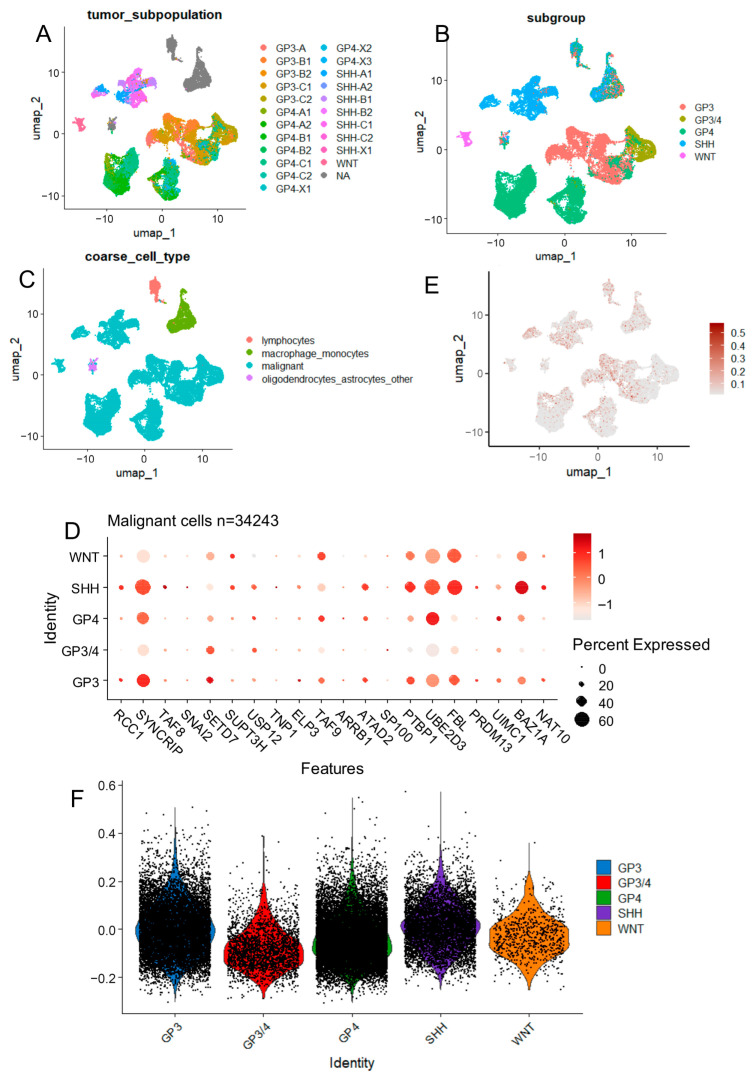
Single-cell distribution of epifactor expression scores in SHH medulloblastoma. (**A**) UMAP of all single cells colored by cell type (malignant vs. non-malignant). (**B**) Cell cycle phase distribution of malignant cells. (**C**) Violin plots of epifactor expression scores by cell type. (**D**) UMAP colored by per-cell epifactor expression score; positive values indicate activation of adverse epigenetic program. (**E**) Dot plot showing expression of 20 adverse epifactors across malignant cells stratified by molecular subtype. (**F**) Violin plots comparing epifactor scores across molecular subtypes of malignant cells.

**Figure 7 cancers-17-03424-f007:**
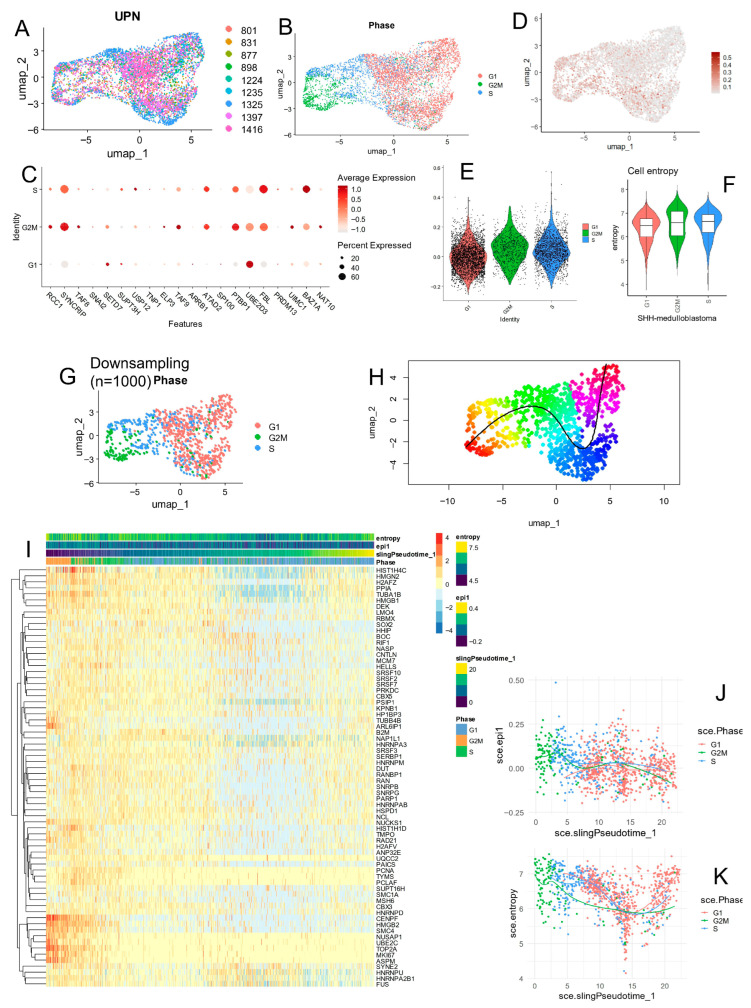
Epifactor score signature is enriched in proliferative G2M SHH malignant cells having high entropy expression program. (**A**) UMAP dimension reduction on malignant cells from nine SHH medulloblastoma patients. (**B**) Phase cycle phase prediction on malignant cells from SHH medulloblastoma samples. (**C**) Epifactor expression score quantification on malignant cells from SHH medulloblastoma. (**D**) Dot plot of adverse epifactors in SHH-MB malignant cells stratified on cell cycle phase prediction. (**E**) Violin plot of epifactor score in SHH-MB malignant cells according to cell cycle phase prediction. (**F**) Violin plot of cell entropy in SHH-MB malignant cells according to cell cycle phase prediction. (**G**) UMAP dimension reduction according to cell cycle phase prediction in SHH-MB malignant cells after downsampling object restricted to one thousand cells. (**H**) Pseudotime quantification cell trajectory prediction in SHH-MB malignant cells. (**I**) Pseudotime expression heatmap of 70 best markers identified on SHH-MB malignant cells. (**J**) Pseudotime expression plot for epifactor score stratified on cell cycle phase prediction. (**K**) Pseudotime expression plot for cell entropy stratified on cell cycle phase prediction.

**Figure 8 cancers-17-03424-f008:**
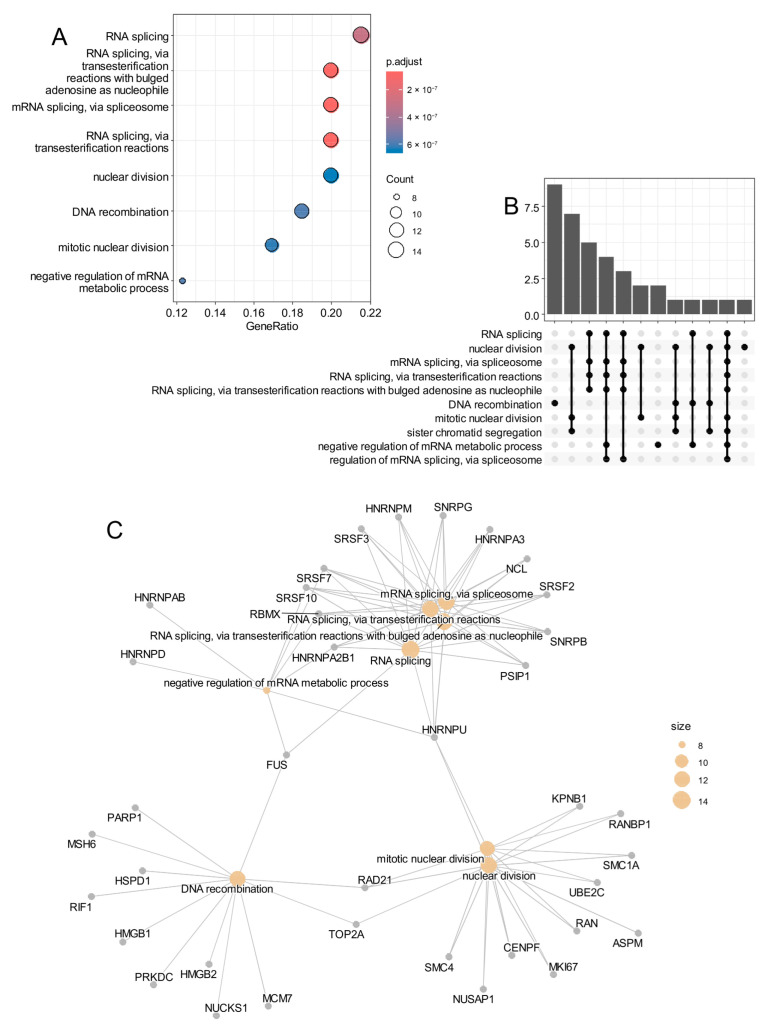
Epifactor cell trajectory in proliferative SHH-MB malignant cells implicated main activation of RNA processing. (**A**) Dot plot of functional enrichment performed on epifactor cell trajectory signature with Gene Ontology Biological Process. (**B**) Upset plot of functional enrichment performed on epifactor cell trajectory signature. (**C**) Functional enrichment plot drawn for epifactor cell trajectory signature activated in proliferative SHH-MB malignant cells.

**Figure 9 cancers-17-03424-f009:**
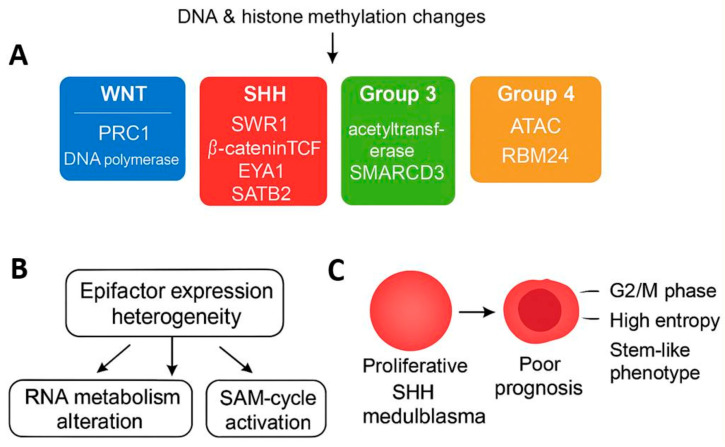
Proposed schematic model summarizing impact of epigenetic factor heterogeneity on medulloblastoma subtypes and prognosis. (**A**) DNA and histone methylation changes drive subtype-specific de-regulation of epigenetic complexes. SWI/SNF alterations occur broadly across all groups; acetyltransferase and ATAC complexes are enriched in Group 3 and Group 4; SWR1 and β-catenin/TCF complexes characterize SHH; and PRC1/DNA polymerase complexes define WNT. (**B**) These epigenetic disturbances converge on heterogeneity of epifactor expression, leading to altered RNA metabolism and activation of S-adenosyl-L-methionine (SAM) cycle. (**C**) In SHH and Group 3 medulloblastomas, these alterations generate high epifactor expression score associated with proliferative G2/M-phase, high-entropy malignant cells exhibiting stem-like behavior and poor prognosis.

**Table 1 cancers-17-03424-t001:** Clinical parameters of PBTA transcriptome cohort: column “total” corresponds to total patient numbers in cohort; column “low”: patients presenting low epifactor score; column “high”: patients presenting high epifactor score; *p*-values were computed by chi square test (qualitative) or by ttest (quantitative) between low- and high-score patients.

Variable	Level	Low (*n* = 116)	High (*n* = 138)	Total (*n* = 254)	*p*-Value
CANCER_TYPE_DETAILED	Medulloblastoma, WNT-activated	13 (11.2)	8 (5.8)	21 (8.3)	
	Medulloblastoma, SHH-activated	11 (9.5)	63 (45.7)	74 (29.1)	
	Medulloblastoma, Group 4	74 (63.8)	25 (18.1)	99 (39.0)	
	Medulloblastoma, Group 3	18 (15.5)	42 (30.4)	60 (23.6)	<1 × 10^−4^
TUMOR_TYPE	Primary	81 (69.8)	74 (53.6)	155 (61.0)	
	Metastatic	31 (26.7)	53 (38.4)	84 (33.1)	
	Recurrence	2 (1.7)	9 (6.5)	11 (4.3)	
	Deceased	0 (0.0)	1 (0.7)	1 (0.4)	
	Progression	2 (1.7)	1 (0.7)	3 (1.2)	0.039831
EXTENT_OF_TUMOR_RESECTION	Gross/Near total resection	71 (68.3)	68 (58.6)	139 (63.2)	
	Partial resection	22 (21.2)	35 (30.2)	57 (25.9)	
	Unavailable	7 (6.7)	6 (5.2)	13 (5.9)	
	Biopsy only	4 (3.8)	7 (6.0)	11 (5.0)	0.350439
TUMOR_FRACTION	Mean (sd)	0.7 (0.2)	0.7 (0.3)	0.7 (0.3)	0.806818
TUMOR_PLOIDY	Mean (sd)	2.5 (0.8)	2.4 (0.7)	2.5 (0.7)	0.384828
CNS_REGION	Ventricles	9 (7.8)	8 (5.8)	17 (6.7)	
	Posterior fossa	76 (65.5)	89 (65.0)	165 (65.2)	
	Mixed	28 (24.1)	38 (27.7)	66 (26.1)	
	Spine	1 (0.9)	0 (0.0)	1 (0.4)	
	Hemispheric	2 (1.7)	1 (0.7)	3 (1.2)	
	Other	0 (0.0)	1 (0.7)	1 (0.4)	0.667556
SEX	Male	83 (71.6)	72 (52.2)	155 (61.0)	
	Female	33 (28.4)	66 (47.8)	99 (39.0)	0.002485

**Table 2 cancers-17-03424-t002:** Clinical parameters of Williamson transcriptome cohort: column “total” corresponds to total patient numbers in cohort; column “low”: patients presenting low epifactor score; column “high”: patients presenting high epifactor score; *p*-values were computed by chi square test (qualitative) or by test (quantitative) between low- and high-score patients.

Variable	Level	Low (*n* = 195)	High (*n* = 136)	Total (*n* = 331)	*p*-Value
gender	male	123 (63.1)	71 (52.2)	194 (58.6)	
	female	54 (27.7)	47 (34.6)	101 (30.5)	0.1329959
age	>16	3 (1.5)	1 (0.7)	4 (1.2)	
	0 to 3	18 (9.2)	26 (19.1)	44 (13.3)	
	3 to 16	158 (81.0)	88 (64.7)	246 (74.3)	
	over 16	1 (0.5)	4 (2.9)	5 (1.5)	0.0065849
developmental.stage	adult	4 (2.1)	5 (3.7)	9 (2.7)	
	infant	18 (9.2)	26 (19.1)	44 (13.3)	
	child	158 (81.0)	88 (64.7)	246 (74.3)	0.0094959
subgroup	Grp4	126 (64.6)	21 (15.4)	147 (44.4)	
	SHH	11 (5.6)	56 (41.2)	67 (20.2)	
	MB-NOS	8 (4.1)	2 (1.5)	10 (3.0)	
	Grp3	19 (9.7)	44 (32.4)	63 (19.0)	
	WNT	24 (12.3)	7 (5.1)	31 (9.4)	
	Grp3/Grp4	7 (3.6)	6 (4.4)	13 (3.9)	<1 × 10^−4^
desmoplastic.nodular	TRUE	11 (7.7)	12 (12.6)	23 (9.7)	
	FALSE	132 (92.3)	83 (87.4)	215 (90.3)	0.2988058
large.cell.anaplastic	FALSE	133 (93.0)	75 (78.9)	208 (87.4)	
	TRUE	10 (7.0)	20 (21.1)	30 (12.6)	0.0026912
ctnnb1.mutation	FALSE	129 (86.0)	102 (95.3)	231 (89.9)	
	TRUE	21 (14.0)	5 (4.7)	26 (10.1)	0.0254499
tp53.mutation	FALSE	189 (99.5)	120 (89.6)	309 (95.4)	
	TRUE	1 (0.5)	14 (10.4)	15 (4.6)	<1 × 10^−4^
tert.mutation	TRUE	3 (2.1)	15 (15.2)	18 (7.4)	
	FALSE	140 (97.9)	84 (84.8)	224 (92.6)	0.0003766
myc.amplification	FALSE	155 (99.4)	94 (88.7)	249 (95.0)	
	TRUE	1 (0.6)	12 (11.3)	13 (5.0)	0.0002977
mycn.amplification	FALSE	151 (96.8)	90 (84.1)	241 (91.6)	
	TRUE	5 (3.2)	17 (15.9)	22 (8.4)	0.0006199
gfi1.rearrangement	FALSE	191 (97.9)	131 (96.3)	322 (97.3)	
	TRUE	4 (2.1)	5 (3.7)	9 (2.7)	0.5816415
gfi1b.rearrangement	FALSE	192 (98.5)	136 (100.0)	328 (99.1)	
	TRUE	3 (1.5)	0 (0.0)	3 (0.9)	0.3877762
prdm6.rearrangement	FALSE	170 (87.2)	133 (97.8)	303 (91.5)	
	TRUE	25 (12.8)	3 (2.2)	28 (8.5)	0.0013109

**Table 3 cancers-17-03424-t003:** Univariate Cox overall survival analysis for 29 adverse predictive epifactors obtained on expression in PBTA transcriptome cohort.

Identifiers	Beta Coefficients	Hazard Ratios	*p*-Values
MBNL1	2.536	12.629	1.13 × 10^−5^
ENY2	2.427	11.32	9.42 × 10^−5^
NAP1L1	1.801	6.055	2.75 × 10^−4^
NPM1	1.39	4.015	4.65 × 10^−4^
FOXO1	1.667	5.297	6.51 × 10^−4^
NEK6	1.723	5.601	8.14 × 10^−4^
TLE4	0.932	2.538	1.05 × 10^−3^
DDX21	1.92	6.821	2.16 × 10^−3^
ZNF516	0.893	2.443	3.10 × 10^−3^
RCC1	1.791	5.997	3.68 × 10^−3^
SYNCRIP	2.145	8.546	4.85 × 10^−3^
TAF8	1.737	5.679	7.57 × 10^−3^
SNAI2	1.006	2.733	1.29 × 10^−2^
SETD7	1.312	3.714	1.35 × 10^−2^
SUPT3H	1.879	6.548	1.56 × 10^−2^
USP12	2.018	7.52	1.69 × 10^−2^
TNP1	1.311	3.711	1.87 × 10^−2^
ELP3	2.074	7.957	1.95 × 10^−2^
TAF9	2.482	11.963	1.99 × 10^−2^
ARRB1	1.022	2.778	2.07 × 10^−2^
ATAD2	1.282	3.603	2.10 × 10^−2^
SP100	1.032	2.806	2.16 × 10^−2^
PTBP1	1.231	3.425	2.59 × 10^−2^
UBE2D3	2.415	11.188	2.92 × 10^−2^
FBL	0.915	2.497	3.49 × 10^−2^
PRDM13	0.81	2.248	4.54 × 10^−2^
UIMC1	1.264	3.54	4.82 × 10^−2^
BAZ1A	0.962	2.616	4.87 × 10^−2^
NAT10	1.554	4.729	4.95 × 10^−2^

**Table 4 cancers-17-03424-t004:** Multi-variable overall survival model on PBTA cohort including epifactor expression score (score) and other covariates: CI95 (confident interval at 95 percent).

Term	Hazard Ratios	CI95.Low	CI95.High	*p*-Value
score	1.187	1.075	1.31	6.69 × 10^−4^
AGE	0.955	0.912	1.001	5.37 × 10^−2^
SEXMale	0.879	0.536	1.441	6.08 × 10^−1^
TUMOR_TYPE: Deceased	34.982	4.107	297.998	1.14 × 10^−3^
TUMOR_TYPE: metastatic	3.004	1.843	4.896	1.02 × 10^−5^
TUMOR_TYPE: progression	2.47	0.572	10.669	2.26 × 10^−1^
TUMOR_TYPE: recurrence	1.589	0.535	4.714	4.04 × 10^−1^

**Table 5 cancers-17-03424-t005:** Molecular subtype comparisons for epifactor single cell expression score in medulloblastoma malignant cells: each row presents one by one ttest comparison between subtypes.

Comparison	Mean Groupe 1	Mean Groupe 2	*t*-Test *p*-Value
GP4 versus SHH	−0.04	0.015	1.68 × 10^−286^
GP4 versus GP3	−0.04	0.001	1.57 × 10^−195^
GP4 versus WNT	−0.04	−0.026	1.55 × 10^−5^
GP4 versus GP3/4	−0.04	−0.08	5.94 × 10^−94^
SHH versus GP3	0.015	0.001	7.13 × 10^−19^
SHH versus WNT	0.015	−0.026	5.24 × 10^−33^
SHH versus GP3/4	0.015	−0.08	0
GP3 versus WNT	0.001	−0.026	9.79 × 10^−16^
GP3 versus GP3/4	0.001	−0.08	8.47 × 10^−296^
WNT versus GP3/4	−0.026	−0.08	4.69 × 10^−49^

## Data Availability

The original contributions presented in this study are included in the article. Further inquiries can be directed to the corresponding author.

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
