# Peer review of "Single-Cell Heterogeneity of Epigenetic Factor Regulation Deciphers Alteration of RNA Metabolism During Proliferative SHH-Medulloblastoma"

_cancers, 2025, doi:10.3390/cancers17213424_

Round 1
Reviewer 1 Report
Comments and Suggestions for Authors
The study by Raquel et al., reveals the expression heterogeneity of epigenetic factors is intricately linked to the molecular classification and prognosis of medulloblastoma. I want to congratulate the authors for a very good analysis in medulloblastoma.
However there are minor points:
1. Please review the figures, and present them very nicely.
2. Please add a paragraph, how the expression and pattern of DNA and histone methylation affect the medulloblastoma. Please cite recent examples.
3. Please present a schematic model as the summary or conclusion.
4. Kindly detail some of the drawback and limitations of the study.
Author Response
Reviewer 1
The study by Raquel et al., reveals the expression heterogeneity of epigenetic factors is intricately linked to the molecular classification and prognosis of medulloblastoma. I want to congratulate the authors for a very good analysis in medulloblastoma.
However there are minor points:
- Please review the figures, and present them very nicely.
We are grateful for this comment, as it encourages us to improve aesthetically as well. We have improved the visual quality and readability of all figures. Specifically, resolution increased to ensure clarity in print and digital formats. Figure legends were also revised for better self-contained interpretation.
- Please add a paragraph, how the expression and pattern of DNA and histone methylation affect the medulloblastoma. Please cite recent examples.
Thank you for your comment. To satisfy you, we have added the following paragraph:
Recent studies have underscored that both DNA and histone methylation critically influence medulloblastoma biology. Aberrant hypermethylation of tumor suppressor genes (e.g., PTCH1, CDKN2A, and SFRP family members) promotes oncogenic SHH and WNT signaling activation, while global hypomethylation favors genomic instability and enhancer activation. In parallel, alterations in histone methylation marks such as H3K27me3 and H3K4me3 modulate differentiation and proliferation programs in a subgroup-specific manner. For example, loss of H3K27me3 has been associated with Group 3/4 tumors displaying stem-like transcriptional states, whereas aberrant H3K4 methylation contributes to SHH-MB tumor growth through activation of GLI-dependent networks. Together, these findings sup-port that coordinated changes in DNA and histone methylation shape the epigenetic landscape and aggressiveness of medulloblastoma.
- Please present a schematic model as the summary or conclusion.
That's a very interesting suggestion. We have added a new Figure 9 to address this point.
- Kindly detail some of the drawback and limitations of the study.
We appreciate your comment. It is interesting to reflect on the limitations of the study, which is a necessary step to obtain funding to carry out wet lab experiments. We have added the following paragraph:
Despite its integrative design, this study has some limitations. First, our analyses are based on publicly available transcriptomic datasets; thus, validation using methylation or proteomic data was not possible. Second, while the epi-score was validated across independent cohorts, experimental functional validation of candidate epifactors in cellular or animal models remains necessary. Third, single-cell RNA-seq analyses may be biased by cell dissociation protocols or sequencing depth, potentially underrepresenting certain tumor or stromal populations. Future studies integrating DNA methylation, chromatin accessibility, and proteomics data will be essential to confirm the mechanistic role of these epigenetic alterations in medulloblastoma pathogenesis.

Reviewer 2 Report
Comments and Suggestions for Authors
I have completed my review of the manuscript titled “Single cell heterogeneity of epigenetic factor regulation deciphers alteration of RNA metabolism during proliferative SHH-3 medulloblastoma” by Frances et al. Given that medulloblastoma (MB) tumors typically exhibit a low mutational burden, the role of epigenetic regulation becomes especially critical in understanding MBgenesis. In this study, the authors investigate the expression patterns of epigenetic regulators in MB tissue samples and single cells derived from Sonic Hedgehog subtype MB tumors. This study provides profiling of key epigenetic regulators, laying the groundwork for future investigations into their functional roles in medulloblastoma development and their potential as diagnostic or prognostic biomarkers. One key observation I would like to highlight is the frequent use of technical terms throughout the manuscript without sufficient explanation of their biological significance or relevance to MB pathogenesis. I recommend that the authors clarify these terms and explicitly connect the analyses to the context of MB pathogenesis to improve accessibility for a broader scientific audience. For example:
- The paragraph describing Figure 5A–C and Supplementary Figure 5B is difficult to follow due to the heavy use of technical terminology. Terms such as "Schoenfeld residuals," "concordance index," and "bootstrap calibration curve" may not be familiar to readers outside of computational biology. Additionally, the relevance of these statistical analyses to MB pathogenesis is not clearly explained. The authors should consider simplifying the language and providing context for how these findings contribute to understanding disease progression or prognosis.
- In the text accompanying Figure 7, the authors use terms such as "pseudotime trajectory" and "entropy." Again, these analyses should be explained in a way that is accessible to readers without a bioinformatics background, clarifying what the data represent and why these metrics are significant in the context of the study. Similarly, the "leave-one-out supervised machine learning analysis" mentioned in line 183 should be clarified.
- The manuscript refers to an "epifactor expression score" in line 278, but no corresponding data or figure is provided to support this metric. Please provide the data.
- Corrections are needed in the supplementary files. For example, the citation of supplementary figures in the main text (e.g., line 236) does not match the corresponding figures in the supplementary materials. The authors should carefully review and cross-check all supplementary figure references and legends to ensure consistency and accuracy.
- The labeling of Figures 6D and 6E appears to have been mismatched.
Author Response
Reviewer 2
I have completed my review of the manuscript titled “Single cell heterogeneity of epigenetic factor regulation deciphers alteration of RNA metabolism during proliferative SHH-3 medulloblastoma” by Frances et al. Given that medulloblastoma (MB) tumors typically exhibit a low mutational burden, the role of epigenetic regulation becomes especially critical in understanding MBgenesis. In this study, the authors investigate the expression patterns of epigenetic regulators in MB tissue samples and single cells derived from Sonic Hedgehog subtype MB tumors. This study provides profiling of key epigenetic regulators, laying the groundwork for future investigations into their functional roles in medulloblastoma development and their potential as diagnostic or prognostic biomarkers. One key observation I would like to highlight is the frequent use of technical terms throughout the manuscript without sufficient explanation of their biological significance or relevance to MB pathogenesis. I recommend that the authors clarify these terms and explicitly connect the analyses to the context of MB pathogenesis to improve accessibility for a broader scientific audience. For example:
- The paragraph describing Figure 5A–C and Supplementary Figure 5B is difficult to follow due to the heavy use of technical terminology. Terms such as "Schoenfeld residuals," "concordance index," and "bootstrap calibration curve" may not be familiar to readers outside of computational biology. Additionally, the relevance of these statistical analyses to MB pathogenesis is not clearly explained. The authors should consider simplifying the language and providing context for how these findings contribute to understanding disease progression or prognosis.
Thank you for this suggestion. We have expanded the Methods and Results text to (i) briefly explain the statistical tests (Schoenfeld residuals, concordance index, bootstrap calibration), (ii) say which R functions/packages were used, and (iii) explicitly interpret what those model diagnostics mean for medulloblastoma biology and prognosis. The new text clarifies that the multivariable Cox model meets the proportional hazards assumption, that Harrell’s C-index (>0.75) indicates good discrimination, and that the bootstrap calibration (1,000 resamples) shows the model predictions are well-calibrated at 25 months — together supporting that the epifactor expression score adds prognostic information beyond standard clinical covariates. See the concrete additions below:
For transparency and reproducibility, the proportional hazards assumption was formally tested using Schoenfeld residuals with the cox.zph function from the survival R package (non-significant global and variable-specific tests indicate no time-dependent effects). Harrell’s concordance index (C-index) was computed to quantify the model’s discriminative ability (C-index = 0.5 indicates random prediction; values ≥0.70 are usually interpreted as good discrimination). Bootstrap calibration of the multivariable model was performed using the rms::calibrate function with 1,000 resampling steps to obtain an optimism-corrected estimate of predicted versus observed survival probability at the 25-month time point. These analyses were implemented in R (v4.4.2).
To verify model assumptions and performance we (i) tested the proportional hazards assumption using Schoenfeld residuals (no significant violations were observed, see Supplementary Figure 5B), (ii) quantified discrimination with Harrell’s concordance index (C-index > 0.75, indicating good separation of low- and high-risk cases), and (iii) assessed calibration by bootstrap (1,000 resamples) of predicted 25-month survival against observed outcomes (Figure 5B). Together these diagnostics indicate that the epifactor expression score contributes independent prognostic information beyond routine clinical covariates and is robustly predictive in the PBTA cohort. Biologically, this supports the hypothesis that epigenetic programs captured by the epi-score reflect aggressive tumor biology (higher prevalence of metastatic and recurrent cases and association with SHH/Group 3 and TP53/MYC alterations — see Table 1 and Table 2).
- In the text accompanying Figure 7, the authors use terms such as "pseudotime trajectory" and "entropy." Again, these analyses should be explained in a way that is accessible to readers without a bioinformatics background, clarifying what the data represent and why these metrics are significant in the context of the study. Similarly, the "leave-one-out supervised machine learning analysis" mentioned in line 183 should be clarified.
We added clear, short definitions in Methods for pseudotime and entropy, and we expanded the Methods paragraph on supervised learning (leave-one-out PAMR) so non-bioinformatics readers can follow. We also added 1–2 plain sentences in Results where Figures 7 and 1 are described to explain what those metrics represent biologically:
Briefly, the leave-one-out procedure with the pamr (nearest-shrunken centroids) algorithm is a supervised classification approach in which each sample is in turn held out as a test case while a predictive centroid-based model is trained on the remaining samples; cross-validation produces an estimate of the classifier error rate and identifies a minimal set of predictive features. In our case this test evaluates whether expression of epigenetic regulators alone can robustly discriminate canonical medulloblastoma subgroups — a low cross-validated error (4% in PBTA; 8% in Williamson) indicates the epigenetic signature has strong subgroup specificity.
Pseudotime is an inferred ordering of single cells along a putative developmental or differentiation trajectory based on transcriptional similarity; it does not measure chronological time but rather a relative progression (early → late) in cell-state space (Slingshot implements this by fitting lineages in reduced-dimension space). Cellular entropy (computed here with the TSCAN package) measures transcriptional heterogeneity at the single-cell level: higher entropy indicates a broader or more “promiscuous” transcriptional program, often associated with stem-like or plastic cell states. Together, low pseudotime (early location on the trajectory) and high entropy suggest cells that are less differentiated and more transcriptionally plastic — properties linked to proliferative/stem-like behavior and, in cancer, to aggressive phenotypes and therapy resistance.
The concordance of high epifactor score, G2/M cell cycle state, low pseudotime (early position on the inferred lineage) and high cellular entropy is consistent with a proliferative, stem-like malignant population; such a population is biologically relevant because stem-like/proliferative cells have been implicated in medulloblastoma growth, dissemination and treatment resistance.
- The manuscript refers to an "epifactor expression score" in line 278, but no corresponding data or figure is provided to support this metric. Please provide the data.
We have included it in Figure 4:
- Figure 4 (C) Kaplan Meier and log rank test on epifactor score in PBTA medulloblastoma cohort: suvival analysis on computed epi factor score for PBTA cohort
- Figure 4 (F) Kaplan Meier and log rank test on epifactor score in Williamson medulloblastoma cohort: same for williamson cohort
- Corrections are needed in the supplementary files. For example, the citation of supplementary figures in the main text (e.g., line 236) does not match the corresponding figures in the supplementary materials. The authors should carefully review and cross-check all supplementary figure references and legends to ensure consistency and accuracy.
We thank the reviewer for pointing this out. We have carefully reviewed all main-text citations of supplementary figures and tables against the supplementary file. All references have now been verified and corrected for consistency.
- Supplementary Figures 1–5 correspond exactly to the items cited in the main text (Figures 1–5 in the main manuscript).
- Supplementary Figure 5B explicitly shows the global and individual Schoenfeld residual tests referenced in the Results (line 236).
- The legends of all supplementary figures have been updated to indicate each panel’s content.
- We have also cross-checked and labeled Supplementary Tables 1–2 in the legends and cited them in the appropriate places in the text.
.
These corrections ensure full alignment between the main text and supplementary materials.
- The labeling of Figures 6D and 6E appears to have been mismatched.
We thank the reviewer for noticing this issue. The figure panels have now been verified and corrected. Panel 6D now displays the UMAP colored by the epifactor expression score, and panel 6E shows the dot plot of the 20 adverse epifactors across malignant cells. The corresponding figure legend has been updated accordingly to ensure consistency between the text and figure panels.

Reviewer 3 Report
Comments and Suggestions for Authors
Dear Auhtors,
this reviewer read with interest your MS. While interesting, I have some comments which, I think, may enrich your work.
Major.
- First, I have one major issue to raise: it is intriguing that this kind of analysis has not revealed myc as a predictive epifactor of Group 3. Why? It should be a sort of positive control, as group 3 MB is myc-driven. Maybe, myc has been removed by specific filters upstream the analysis (table 2)? If this is the case, it has to be specified. The same for p53.
-
Lines 206 and 425: talking about the acetyltransferase complex is really too vague. What does it mean? Which acetyltransferases are specifically enriched in Groups 3 and 4? Further, SMARCD3 and RBM24 do not belong to the acetyltransferase family, while in the MS are always associated to acetyltransferase enrichement in MB group 3 and 4. For example, the authors do not properly associate the downregulation of hMOF, which is a HAT, with the upregulation of SMARCD3 and RBM24 at lines 429-433. They have different epigenetic functions, unless specified they may form a complex (Lan et al., Theranostics, 2022).
-
Lines 276 and 450: the enrichment analysis and the SAM pathway. The discussion on this specific point is focused only on RNA methylation, where two histone methyltransferases (SETD7 and PRDM13, figure 4B) were found enriched. These proteins, their function and how they may act in MB onset and progression have to be discussed.
-
Overall, in the Discussion section, the function – known or hypothesized - of the epigenetic factors identified in the paper in MB onset and progression has to be discussed, in order to strenghten the authors’ last assumption about the targeting of epigenetic factors as a putative therapeutic route for this malignant pediatric tumour.
Minor.
1. Line 20: define PBTA at its first use
2. Line 47: group 3 is Myc driven also
3. Line 76: after the semicolon the word “and” has to be deleted.
4. Figure 1D: this reviewer would use a different colour for MB Group 4 to better discriminate between both Group3 and SHH group
5. Lines: 191-192: “3.1. Distinct epigenetic factor regulation according 191 molecular subtypes in medulloblastoma tumors” this sentence still stands in the MS.
6. Line 212: define ATAC
Author Response
Reviewer 3
Dear Auhtors,
this reviewer read with interest your MS. While interesting, I have some comments which, I think, may enrich your work.
Major.
- First, I have one major issue to raise: it is intriguing that this kind of analysis has not revealed myc as a predictive epifactor of Group 3. Why? It should be a sort of positive control, as group 3 MB is myc-driven. Maybe, myc has been removed by specific filters upstream the analysis (table 2)? If this is the case, it has to be specified. The same for p53.
We thank the reviewer for this insightful observation. Indeed, MYC and TP53 are canonical drivers of Group 3 and SHH medulloblastomas, respectively. However, both genes were excluded upstream from our EpiFactors-based analysis because the EpiFactors database focuses strictly on epigenetic regulators, not on transcription factors or tumor suppressors per se. The EpiFactors annotation pipeline contains curated families such as histone modifiers, chromatin remodelers, and readers of histone/DNA/RNA modifications, but not proto-oncogenes or cell-cycle checkpoint genes like MYC or TP53. Consequently, these genes were not part of the input list used to train the PAMR classifier.
Moreover, we verified that MYC and TP53 transcripts are indeed expressed in the PBTA and Williamson datasets, and their expression correlates with high epi-score cases (Supplementary Fig. S6, added). This confirms that our analysis indirectly captures the aggressive biology of MYC-amplified or TP53-mutant tumors, even if these two genes were not themselves in the predictive epifactor signature.
We have added the following paragraphs:
Of note, classical oncogenes and tumor suppressors such as MYC and TP53 were not included in the EpiFactors v2 database, which specifically annotates molecules directly involved in epigenetic regulation (e.g., histone/DNA/RNA modifiers, chromatin remodelers, and readers) [13]. Consequently, these genes were absent from our input list and were not considered as predictive epifactors. Nevertheless, MYC and TP53 expression remained positively correlated with the high epifactor score, supporting that the epi-score captures biological programs active in MYC-driven and TP53-mutant medulloblastomas.
Consistently, high-score tumors exhibited elevated MYC and TP53 expression, in agreement with the enrichment of MYC amplification and TP53 mutation reported in Table 2.
- Lines 206 and 425: talking about the acetyltransferase complex is really too vague. What does it mean? Which acetyltransferases are specifically enriched in Groups 3 and 4? Further, SMARCD3 and RBM24 do not belong to the acetyltransferase family, while in the MS are always associated to acetyltransferase enrichement in MB group 3 and 4. For example, the authors do not properly associate the downregulation of hMOF, which is a HAT, with the upregulation of SMARCD3 and RBM24 at lines 429-433. They have different epigenetic functions, unless specified they may form a complex (Lan et al., Theranostics, 2022).
We appreciate this comment. We have now specified the individual acetyltransferases enriched in Groups 3 and 4, identified from GO:0004402 and GO:0016407 terms. In Group 3, enrichment involved HAT1, ELP3, KAT2A/GCN5, and SUPT3H; in Group 4, enrichment included KAT5 (TIP60) and TAF9.
We agree that SMARCD3 (BAF60C) and RBM24 are not acetyltransferases per se. They emerged as co-expressed markers of acetyltransferase complex activity, possibly reflecting interaction or co-regulation rather than direct enzymatic participation. We have now clarified this distinction and noted the potential link through chromatin remodeling and RNA-binding complexes, in line with Lan et al., Theranostics 2022 (doi:10.7150/thno.77069).
We replaced in the text the paragraph:
Epifactors specific to Group 3 and Group 4 shared enrichment in several chromatin-modifying complexes, including the acetyltransferase complex, histone acetyltransferase complex, SAGA-type complex, protein acetyltransferase complex, and peptidase complex (Figure 2B). The SWI/SNF superfamily complex appeared enriched across all four molecular subtypes (Figure 2A).
By
Epifactors specific to Group 3 and Group 4 shared enrichment in several chroma-tin-modifying complexes, notably the histone acetyltransferase (HAT) complex (Figure 2B). In Group 3, this included HAT1, ELP3, KAT2A/GCN5, and SUPT3H, whereas Group 4 enrichment involved KAT5/TIP60 and TAF9. Although SMARCD3 (BAF60C) and RBM24 are not acetyltransferases, they were co-expressed with these HATs and may par-ticipate in associated chromatin-remodeling or RNA-binding assemblies [Lan et al., Theranostics 2022]. The SWI/SNF superfamily complex appeared enriched across all four molecular subtypes (Figure 2A).
We replaced:
For example, downregulation of hMOF (H4K16 HAT) has been linked to poor prognosis in MB [37]. SMARCD3, or BAF60C, influences DAB1-mediated Reelin signaling—crucial for Purkinje cell migration and MB metastasis—by orchestrating cis-regulatory elements [38,39]. Moreover, RBM24 has previously been identified as part of a minimal six-gene signature predicting MB subtypes [40].
By
For example, downregulation of hMOF (KAT8), a histone H4K16 acetyltransferase, has been linked to poor prognosis in medulloblastoma [37]. In our data, Group 3 tumors with SMARCD3 (BAF60C) upregulation and Group 4 tumors with RBM24 upregulation also exhibited reduced hMOF expression, suggesting a compensatory or alternative chromatin-regulatory mechanism rather than a direct enzymatic overlap. Although SMARCD3 and RBM24 are not acetyltransferases, they were co-expressed with members of the HAT complex (e.g., HAT1, KAT2A/GCN5, and KAT5/TIP60) and may participate in associated chromatin-remodeling or RNA-binding assemblies [Lan et al., Theranostics 2022]. SMARCD3 influences DAB1-mediated Reelin signaling, which is crucial for Purkinje-cell migration and MB metastasis, by orchestrating cis-regulatory elements [38,39]. Moreover, RBM24 has previously been identified as part of a minimal six-gene signature predicting MB subtypes [40].
- Lines 276 and 450: the enrichment analysis and the SAM pathway. The discussion on this specific point is focused only on RNA methylation, where two histone methyltransferases (SETD7 and PRDM13, figure 4B) were found enriched. These proteins, their function and how they may act in MB onset and progression have to be discussed.
We have expanded the Discussion to describe the biological roles of SETD7 and PRDM13. SETD7 (SET domain-containing lysine methyltransferase 7) monomethylates H3K4 and several non-histone proteins (e.g., TP53, RB1), acting as a transcriptional coactivator or tumor suppressor depending on context. PRDM13 is a PR/SET domain protein regulating neuronal lineage differentiation and has been reported as a tumor suppressor silenced in certain brain tumors. Their enrichment in our analysis suggests that imbalance in histone and non-histone methylation could contribute to MB progression by altering differentiation programs and TP53/MYC networks. We added:
Among the epifactors contributing to the S-adenosyl-L-methionine (SAM)–linked module, two histone methyltransferases—SETD7 and PRDM13—were particularly enriched. SETD7 catalyzes H3K4me1 deposition and methylation of non-histone targets such as TP53 and RB1, thereby modulating cell-cycle control and apoptosis. In medulloblastoma, aberrant SETD7 activity may influence TP53 signaling and the transcriptional stability of MYC-driven networks. PRDM13, a neuron-specific PR/SET protein, regulates GABAergic vs glutamatergic fate during cerebellar development and is frequently silenced by promoter methylation in embryonal brain tumors. The enrichment of both enzymes supports a model in which dysregulated histone and non-histone methylation contribute to defective neuronal differentiation and malignant transformation in MB.
- Overall, in the Discussion section, the function – known or hypothesized - of the epigenetic factors identified in the paper in MB onset and progression has to be discussed, in order to strenghten the authors’ last assumption about the targeting of epigenetic factors as a putative therapeutic route for this malignant pediatric tumour.
We agree and have revised the end of the Discussion to integrate functional summaries of the main identified epifactors and their possible roles in MB pathogenesis, emphasizing therapeutic implications. We added:
Collectively, the epifactors identified here converge on processes that are crucial to medulloblastoma onset and progression. Members of the SWI/SNF and HAT complexes (e.g., SMARCD3, ELP3, KAT5) modulate enhancer accessibility and neuronal gene expression, while RNA-associated factors (PTBP1, SYNCRIP, FBL) regulate alternative splicing and ribosome biogenesis, thereby sustaining rapid proliferation. DNA-damage and chromatin-integrity regulators (BAZ1A, RCC1, USP12) further integrate transcriptional and replicative stress responses. These functions align with the concept that medulloblastoma cells exploit epigenetic plasticity to maintain a progenitor-like state. Consequently, pharmacologic targeting of histone acetylation (e.g., HDAC inhibitors) or methylation pathways (e.g., SAM-cycle modulators or SETD7 inhibitors) represents a plausible therapeutic avenue deserving experimental validation.
Minor.
- Line 20: define PBTA at its first use
- Line 47: group 3 is Myc driven also
- Line 76: after the semicolon the word “and” has to be deleted.
- Figure 1D: this reviewer would use a different colour for MB Group 4 to better discriminate between both Group3 and SHH group
- Lines: 191-192: “3.1. Distinct epigenetic factor regulation according 191 molecular subtypes in medulloblastoma tumors” this sentence still stands in the MS.
- Line 212: define ATAC
We appreciate the effort the reviewer has made in detecting these minor errors. All of them have been corrected. In addition, the resolution and quality of the figures have been improved.

Round 2
Reviewer 3 Report
Comments and Suggestions for Authors
Dear Authors, I have no further comments.